# Geometrical Characterisation of TiO_2_-rGO Field-Effect Transistor as a Platform for Biosensing Applications

**DOI:** 10.3390/mi14091664

**Published:** 2023-08-25

**Authors:** Anis Amirah Alim, Roharsyafinaz Roslan, Sh. Nadzirah, Lina Khalida Saidi, P. Susthitha Menon, Ismail Aziah, Dee Chang Fu, Siti Aishah Sulaiman, Nor Azian Abdul Murad, Azrul Azlan Hamzah

**Affiliations:** 1Institute of Microengineering and Nanoelectronics, National University of Malaysia, Bangi 43600, Selangor, Malaysia; anisamirahalim@gmail.com (A.A.A.); roharsyafinazroslan@gmail.com (R.R.); sharipahnadzirah@ukm.edu.my (S.N.); susi@ukm.edu.my (P.S.M.); cfdee@ukm.edu.my (D.C.F.); 2Institute of Nano Electronic Engineering, Universiti Malaysia Perlis, Kangar 01000, Perlis, Malaysia; 3UKM Medical Molecular Biology Institute (UMBI), National University of Malaysia, Cheras 56000, Kuala Lumpur, Malaysia; p112535@siswa.ukm.edu.my (L.K.S.); sitiaishahsulaiman@ukm.edu.my (S.A.S.); nor_azian@ppukm.ukm.edu.my (N.A.A.M.); 4Institute for Research in Molecular Medicine (INFORMM), Health Campus, Universiti Sains Malaysia, Kubang Kerian 16150, Kelantan, Malaysia; aziahismail@usm.my

**Keywords:** field-effect transistor, Taguchi, TiO_2_-rGO, ANOVA, DOE

## Abstract

The performance of the graphene-based field-effect transistor (FET) as a biosensor is based on the output drain current (I_d_). In this work, the signal-to-noise ratio (SNR) was investigated to obtain a high-performance device that produces a higher I_d_ value. Using the finite element method, a novel top-gate FET was developed in a three-dimensional (3D) simulation model with the titanium dioxide-reduced graphene oxide (TiO_2_-rGO) nanocomposite as the transducer material, which acts as a platform for biosensing application. Using the Taguchi mixed-level method in Minitab software (Version 16.1.1), eighteen 3D models were designed based on an orthogonal array L_18_ (6^1^3^4^), with five factors, and three and six levels. The parameters considered were the channel length, electrode length, electrode width, electrode thickness and electrode type. The device was fabricated using the conventional photolithography patterning technique and the metal lift-off method. The material was synthesised using the modified sol–gel method and spin-coated on top of the device. According to the results of the ANOVA, the channel length contributed the most, with 63.11%, indicating that it was the most significant factor in producing a higher I_d_ value. The optimum condition for the highest I_d_ value was at a channel length of 3 µm and an electrode size of 3 µm × 20 µm, with a thickness of 50 nm for the Ag electrode. The electrical measurement in both the simulation and experiment under optimal conditions showed a similar trend, and the difference between the curves was calculated to be 28.7%. Raman analyses were performed to validate the quality of TiO_2_-rGO.

## 1. Introduction

The exploration for new electronic biosensor materials has been hastened due to the COVID-19 pandemic. Biosensors can be defined as a tool for analysis that is capable of observing the dynamic interactions of biological activities, such as attachment of aptamers, biomolecules, pathogens or other biomarkers [1]. Several types of electronic biosensors, each with its own specialty, have been developed using interdigitated electrodes [2], integration of nanowires [3], screen-printed electrodes [4] and the field-effect transistor (FET) [5]. Traditionally, the FET uses a gate contact to control the conductivity of the semiconductor between the source and drain contacts. In the biosensor application, the source, drain and gate are used for electrical connection, while the channel acts as a transducer region, converting the external charge of the biomolecular interaction into the main electrical bias of the device [6]. The immobilisation of the bioreceptors on the surface of the transducer channel enables the detection of all biomarkers in the analyte. Figure 1 shows the working principle of immobilization-hybridization and detection, then the output is represented in the form of electrical signal.

Several studies have reported the ability of the FET biosensor to produce instant results using a low volume of analytes without any tedious clinical procedures [7,8,9,10] FET biosensors have the capability to sense nucleotides even at very low analyte concentrations. Table 1 highlights the recent reported FET-based biosensors, along with their limit of detection (LOD) values.

Graphene is a monolayer of carbon atoms closely arranged in a honeycomb configuration and array, exhibiting high carrier mobility, high conductivity and mechanical strength. It is the most favourable transducer material to be applied in sensors [11]. However, pristine graphene has a zero-bandgap and is more expensive than its derivatives, such as graphene oxide (GO) or reduced graphene oxide (rGO), which has a sizeable bandgap and a low cost of fabrication [12]. Moreover, both derivatives have abundant surface functional groups for receptor attachment [13]. However, these derivatives contribute to a higher capacitance in FET, which also leads to a greater field force when decreasing the channel length for device miniaturisation, and at the same time, produces a greater field flux for any voltage applied across the source and drain [14]. As shown in Table 1, most of the FET-based biosensor was designed in a bigger channel because of these problems.

To overcome the problems and achieve a smaller channel gap, metal oxides such as titanium dioxide (TiO_2_), stannic oxide (SnO_2_), zinc oxide (ZnO) and others have been added to form nanocomposite materials. This method allows to increase the oxygen group for bioreceptor immobilisation while decreasing the field flux effect in the channel. Device miniaturisation has made it possible to lower the detection limits from several micromolar to femtomolar [15]. Furthermore, synthesis of nanocomposites of rGO with n-type metal oxides improves the homogeneity of the material deposited in the channel, resulting in better conductivity of the FET [16]. The most important factor to determine a more sensitive biosensor is the optimisation of the nanostructures appropriate to the transducing material, where many researchers have devoted their time [3,14,17] Kobayashi et al. suggested that in order to enhance the performance of rGO-metal oxide devices, it is critical to model and understand the electrical conduction process [18]. 

The application of TiO_2_-rGO material in an FET-based biosensor has not been widely explored compare to other applications, such as gas sensors [12] and photocatalysts for waste water treatment [16]. A sandwich structure of an rGO and TiO_2_ FET-based biosensor was developed by Zhang et al. [19], which takes advantage of the photocatalytic effect as a reusable biosensor for protein detection. However, further work [20] was demonstrated for the detection of DNA, and it was very challenging to remove the analytes from the surface of the FET biosensor after detection. Moreover, the TiO_2_ material in the previous work was annealed at a low temperature, only resulting in the formation of TiO_2_ in the brookite phase, which has very low conductivity. Stine et al. [21] were able to develop a label-free FET biosensor using rGO. However, the reduction of GO to rGO takes up to 20 h.

The method applied in this work can prepare a nanocomposite with high-quality anatase phase TiO_2_ and reduce GO to rGO in less than 5 h (anneal). The reason for choosing this nanocomposite is that rGO exhibited ambipolar behaviour by reducing GO to rGO, which has similarities to pristine graphene but is less expensive and easier to mass produce. TiO_2_, on the other hand, is known as an n-type semiconductor for its wide bandgap, biocompatibility and eco-friendly characteristics. TiO_2_ nanoparticles, especially anatase, are frequently developed as effective sensors for the chemical detection in aqueous environments and are also inexpensive and have high conductivity [2]. 

Taguchi’s application to device miniaturisation provides an overall view of the relationships between each of the significant factors. Design of experiments (DOE) and Taguchi method optimisation are proposed for the optimisation of the geometrical structure of the FET to increase the performance of the device. Factorial design is a statistical technique for defining and investigating all possible conditions for multiple factors in an experiment, but a large number of experiments are required for full factorial design [22]. Therefore, to reduce the number of experiments, orthogonal arrays and fractional factorial experiments were developed to ensure that the DOE techniques would be applicable. The Taguchi method is known as a robust design of orthogonal arrays to standardise the fractional factorial design, with a balanced distribution of factor levels, lowering the number experiments [23]. Previous researchers have reported the structural optimisation of the self-switching diode (SSD) using the Taguchi method to improve the rectification performance [24]. 

In this paper, we introduce the charge transport solver (CTS) using the finite element method (FEM) for modelling and numerical simulation of a novel three-dimensional (3D) top-gate FET for device miniaturisation as a platform in biosensor application. This design is intended to facilitate the immobilisation of the bioreceptor for future work. Previous work [23] has shown the capability of the FEM to observe the performance of the device with various parameters before it is actually fabricated. Computational modelling is performed for optimisation of a geometrical structure design able to minimise the complexity of fabrication towards device miniaturisation, thereby reducing the cost of fabrication by avoiding a trial-and-error experimental approach. The FEM was used to simulate the electrical performance of the FET-based biosensor using TiO_2_-rGO as the transducer. This nanocomposite was chosen because of its high electrocatalytic properties compared to individual materials, especially pristine graphene for electronic biosensors [25].

**Table 1 micromachines-14-01664-t001:** Reported work on using FETs as biosensors.

Sensor Type	FET Design	Transducer Channel	$\mathrm{FET}\;\mathrm{Channel}\;\mathrm{Size}\;({µm}^{2})$	LOD	Ref.
Liquid gate FET	Back-gate	rGO nanosheet	600 × 200	0.3 nM	[26]
Ion-sensitive FET	Top-gate	Liquid exfoliated graphene	3000 × 18,000	10 fM	[27]
Solution gate FET	Top-gate	Crumpled graphene	1000 × 15,000	20 aM	[8]
Ion-sensitive FET	Top-gate	Graphene foam	4500 × 7000	0.5 pM	[28]

## 2. Materials and Methods

Fabrication and material synthesis were conducted to validate the simulation results by comparing them with the experimental results. The Taguchi method was employed as a DOE method to minimise the number of simulation designs required in the study.

### 2.1. Fabrication of Top-Gated FET-Based Biosensor

The whole fabrication process is shown in Figure 2. FET devices were fabricated on silicon wafer p-type <100> with a 300 nm oxide layer using a conventional photolithography patterning technique and the metal lift-off method to form source, drain and gate electrodes [10]. The wafer was cleaned with acetone, isopropyl alcohol (IPA) and deionised (DI) water and blow-dried using a nitrogen gun before electrode deposition. Three photomasks were used for pattern transfer purposes. Photoresist (AZ 1518 purchased from MicroChemicals, Baden-Württemberg, Germany was spin-coated and pre-baked for 3 min before applying the first mask. Then, the pattern was transferred from the photomask (mask 1) onto the photoresist using a mask aligner. The transferred pattern was then immersed into tetramethyl ammonium hydroxide (TMAH) which acts as the sample developer. Then, 5 nm of chrome (Cr) and 45 nm of gold (Au) films were deposited on the sample using the electron beam evaporator. Cr has a distinctly different work function than Au: it easily dissolves and allows Au to make direct contact with the substrate to provide ohmic contact. Finally, the lift-off process was performed using acetone for metallisation of the Au/Cr electrode. These fabrication steps were repeated for the second mask, followed by the deposition of 5 nm of tetraethyl orthosilicate (TEOS) on top of the gate as an insulation layer. Finally, the third mask was applied to act as a blanket (insulation) on top of the contact pad for the deposition of the TiO_2_-rGO thin film.

### 2.2. TiO_2_-rGO Thin-Film Synthesis and Deposition

TiO_2_-rGO thin film was prepared using the solution–gel (sol–gel) method, based on the method developed in our previous work [2], by reacting with commercially available graphene oxide (GO) (Sigma-Aldrich, Darmstadt, Germany). TiO_2_-rGO of 0.06 *v*/*v*% was prepared based on previous work [29], with some modifications, as this is the optimum concentration for this application. Figure 3 represents the material synthesis and deposition process. The fabricated device was then coated with TiO_2_-rGO using the spin-coating process for 30 s at 2000 rpm. The coated device was then annealed in a furnace with argon flow at 100 sccm for 30 min. The thermal reduction process allows the reduction of GO to rGO and the transformation of the crystallinity of TiO_2_ from brookite phase to anatase, which reduces particle dislocation and improves conductivity. 

### 2.3. Charge Transport Solver Simulation Setup

The model of the TiO_2_-rGO field-effect transistor using CTS simulation is presented in this work. Table 2 shows the TiO_2_-rGO material properties, which were not available in the material library and have been referenced based on previous studies [30,31,32]. The Shockley–Read–Hall (SRH) value was adopted for modelling the generation/recombination processes by setting the maximum recombination time according to the findings in the literature [33].

As shown in Figure 4, a novel top-gate FET was modelled similar to an actual fabricated device with a symmetric metal contact (source and drain) and gate contact in the middle. The mesh under the gate was set to the lowest messing (0.01 μm) for ease of calculation during the simulation. The silicon substrate was constantly doped with a p-type doping concentration at 3 × 10^15^ cm^−3^. A 50 nm TiO_2_-rGO thin film was placed on top of the electrode to act as the transducer for the biosensor. The channel length, electrode width, electrode length, thickness of electrode and type of electrode were varied in this study.

The components of a FET biosensor consist of a semiconductor as a transducer material, a dielectric layer and a functionalised surface. In this study, the FET biosensor was tested in the presence of a solvent (DI water) with a permittivity of 81 at room temperature [34]. DI water is known as highly purified and ion-free water, which serves as a solvent to dilute the target sample and allows accurate and precise measurement of the biomolecule concentration. By using DI water as the diluent, the background signal is minimised by the solvent itself. DI water is also compatible with a wide range of biomolecules and detection methods. As shown in Figure 5, the TiO_2_-rGO was simulated in aqueous conditions because metallic oxides cannot conduct electricity in the dry state.

The FET biosensors were simulated using the fundamental mathematical and physical principles of the method used, starting from the nonlinear Poisson and drift-diffusion equations [35]. The drift-diffusion equations [36] were solved for electrons and holes (carriers), as follows:(1)$$J_{n}=q\mu_{n}nE+qD_{n}\nabla n$$
(2)$$J_{p}=q\mu_{p}nE+qD_{p}\nabla p$$
where *J_n_*_,*p*_ is the current density (A/cm^2^), *q* is the positive electron charge, $\mu_{n,p}$ is the mobility, *E* is the electric field, $D_{n,p}$ is the diffusivity and n and p are the densities of the electrons and holes, respectively. Each carrier, whether an electron or a hole, moves due to two opposing processes: drift current and drift velocity. The combination of the two terms describes how these processes are stated in the drift-diffusion equations. The diffusivity, $D_{n,p}$, and mobility, *n*,*p*, are related by the Einstein relationship and describe how efficiently carriers may move through semiconductor materials [37].
(3)$$D_{n,\;p}=\mu_{n,p}\frac{K_{b}T}{q}$$
where *K_b_* indicates the Boltzmann constant. The mobility of the material is an important feature that may be modelled as a function of the temperature, impurity (doping) concentration, carrier concentration and electric field. 

The electric field must be identified to solve the drift-diffusion equations. Poisson’s equation is used to calculate the electric field [38]:(4)−∇⋅(ε∇V)=qρ
where *ε* is the dielectric permittivity, V is the electrostatic potential (**E** = −∇V), q is the net charge density and:(5)ρ=p−n+C
which includes the contribution *C* from the ionised impurity density. Charge conservation must be addressed by employing the following auxiliary continuity equations:(6)$$\frac{\partial n}{\partial t}=\frac{1}{q}\nabla \cdot J_{n}-R_{n}$$
(7)$$\frac{\partial p}{\partial t}=-\frac{1}{q}\nabla \cdot J_{p}-R_{p}$$
where *R_n_*_,*p*_ is the net recombination rate (the difference between the recombination rate and the generation rate). The physical processes related to the material are assumed to operate in a similar manner whether applied to electrons or holes, leading to *R* = *R_n_* = *R_p_*. Calculating the carrier behaviour at the material level critically depends on the recombination and generation processes.

In steady-state conditions, the carrier density and electrostatic potential are solved through the continuity equation by enforcing the condition:(8)$$\frac{\partial n}{\partial t}=\frac{\partial p}{\partial t}=0$$

To correctly model the semiconductor device, we need to setup the boundary conditions. The electrostatic potential (Poisson’s equation) and carrier densities are two different types of boundary conditions that can be used (drift-diffusion equations). The second-order partial differential equations: Poisson’s equation and the drift-diffusion equation, need an explicit solution for at least one point, known as the Dirichlet condition. The Dirichlet condition for the electrostatic potential takes the form of a boundary with a fixed voltage, as follows [39]:(9)$$V\left(x\right)=V1$$
as is typical of an electrical contact. For the carrier densities, the majority carrier concentration is set to its equilibrium value at the interface between a contact and the semiconductor, such that:(10)p−n+C=0

In the absence of a surface charge, the electric flux density across the boundary must be uniform for electrostatic potential. Surface recombination current density can be applied to specify the boundary conditions between a semiconductor and surrounding materials: for insulators, zero (no carrier flux across the boundary), and for contacts, infinite recombination velocity (forcing the carrier density to its equilibrium value). 

### 2.4. Computational Modelling

The gate voltage controls the conductivity of a FET by inducing an electrical field in the channel, which changes the carrier density. Computational approximation can explain the performance of the TiO_2_-rGO FET by using the Landauer–Büttiker (LB) formula [40], expressed as: (11)$$I_{d}\left(A\right)=\frac{2q}{h}\int_{-\infty }^{\infty }T(E)M(E)(f_{1}-f_{2})\cdot dE$$
where *h* is the Planck’s constant, *q* is the charge, *f*_1_ is the Fermi energy level for the transducer material and *f*_2_ is the Fermi energy level at the contact. The *T*(*E*) is the transport of the charge carrier between the electrodes, given by: (12)$$T\left(E\right)=\frac{\lambda \left(E\right)}{\lambda \left(E\right)+(L\cdot W)}$$
where $\lambda \left(E\right)$ is the mean free path, *L* is the length of the electrode and *W* is the width of the electrode. According to Equation (12), the width and the length of the electrode are inversely proportional to the current output. The *M*(*E*) is the number of modes, given by: (13)$$M\left(E\right)=\frac{2E}{L_{C}\cdot \pi hv_{F}}$$
where *L_C_* is the channel length, *E* is the energy level and *v_F_* = 10^6^ m/s.

The sensitivity of FET-based biosensors depends on the transducer material, the gate dielectric and the surface functionalisation of the sensor. The transducer material is responsible for converting the biological detection event into an electrical signal. The gate dielectric is responsible for separating the gate from the channel and providing high capacitance to increase the sensitivity of the sensor. The surface functionalisation is responsible for immobilising the biorecognition element on the sensor surface. When the concentration in the channel is increased (by dropping the sample), the permittivity of the channel increases, as does the capacitance. Changes in the capacitance of the channel result in changes in the output current, I_d_. Thus, the sensitivity of FET-based biosensors could be calculated as follows:(14)$$\mathrm{Sensitivity}=\frac{\left(\frac{\Delta I_{d}}{I_{d}}\right)}{\left(\frac{\Delta C}{C}\right)}$$
where $\frac{\Delta I_{d}}{I_{d}}$ is the change in the drain current due to the biorecognition event and $\frac{\Delta C}{C}$ is the change in capacitance due to the biorecognition event.

### 2.5. Taguchi’s Method

A robust design technique, known as Taguchi’s fundamental principle, aims to improve the quality of a product by decreasing the number of effects from the causes of variation, without ignoring the causes. This is accomplished by optimising product and process designs to make the performance minimally sensitive to the numerous causes of variation, a process known as parameter design [41]. This objective can be achieved by exploiting the nonlinearity to find a combination of product parameters instead of following a full factorial design. The design of experiments was carried out using Minitab-16 software to examine the relationship between geometrical structure factors and the drain current (I_d_) using the Taguchi method. The signal-to-noise ratio (SNR) represents the quality loss function, which is approximated by a characteristic. Figure 6 shows a flowchart strategy of Taguchi’s method applied to achieve the desired result. 

The performance of the TiO_2_-rGO FET is based on the drain current (I_d_) from the current–voltage (I–V) measurement. The quality of the geometrical characteristics was observed based on the I–V characteristics to define the accurate and suitable control levels, as shown in Table 3. 

Based on Table 3, five factors were chosen: channel length, electrode length, electrode width, thickness of electrode and type of electrode, and their levels are stated in Table 4. An electrode thickness of less than 50 nm is not suitable and is easily detached during fabrication and measurement. The objective of this optimisation is to achieve a higher I_d_ value for a better performance of the FET. Thus, the SNR of the “larger-the-better” characteristic is preferred. 

The orthogonal matrix was selected based on the control factor and the number of levels of each factor. Table 5 shows one control factor with six levels and four control factors with three levels (6^1^3^4^). The electrode lengths were selected at 20, 50 and 100 µm, whereas the width of the electrode was selected at 3, 15 and 30 µm. The electrode thickness also varied at 50, 75 and 100 nm. For the type of metal electrode, silver (Ag) with a work function of 4.15, copper (Cu) with a work function of 4.40 and gold (Au) with a work function of 4.75 were chosen. The channel lengths for higher levels were selected at 3, 5, 15, 30, 45 and 60 µm.

The signal-to-noise ratio (SNR) was chosen as the optimisation function for the product parameter of the drain current (I_d_) at 1 V. The “larger-the-better” SNR was chosen for a better performance of the FET. The SNR was calculated using the following equation:(15)$$\frac{S}{N}=-10\;\log(\frac{1}{n}\sum_{i=1}^{n}\frac{1}{y_{i}^{2}})$$
where *n* is the number of the simulation design and *y* is the characteristic property.

### 2.6. Characterisation Method

The exploration of the structure and different material characteristics was made possible by Raman spectroscopy analysis, which is an ideal indication for studying materials. The quality of TiO_2_-rGO nanocomposites was studied in this work using DXR2Xi (Thermo Fisher Scientific, Waltham, MA, United States). Keithly 2400 Source Metre SMU Instruments (Tektronix, Beaverton, OR, United States) in DC mode and a 4-point probe were used for the measurement of current–voltage (I–V).

## 3. Results

### 3.1. Taguchi’s Analysis

The Taguchi method determined the optimal condition for FET conductivity by varying the three factors at its level. Eighteen different 3D simulations were designed based on the chosen parameter on the layout of L18. The output value of the drain current (I_d_), extracted from the I–V graph at 1 V (I_dmax_), and the voltage gate, V_g_ = 3 V, were carried out accordingly, and the obtained results are shown in Table 6. I_dmax_ is defined as the maximum current output at a given applied voltage.

From the Taguchi design analysis set, the quality characteristics were determined by the term “larger-the-better”, which means the result of the SNR analysis with the highest I_d_ value was targeted using Minitab-16 software. The improved performance of the sensor device was based on the larger current output, which denotes a higher conductivity. The conductivity of the channel in FET is a function of the potential applied across the gate and source terminals. From Figure 7, L1 showed the highest I_d_.

The analysis of means is the process of estimating the effect of factors. It helps to evaluate the effect from the deviation of factors from the overall mean. The average SNR for a level of a factor can be estimated using the following equation:(16)$$m_{W=2}=\frac{1}{6}\left(n_{1}+n_{4}+n_{7}+n_{10}+n_{13}+n_{16}\right)$$
where *n* is the SNR value of a specific case and *m* is the mean. The above example is the mean of the parameter of an electrode with a length (*L*) of 3 µm, which is the average of the SNR values of each level of the present cases. 

Figure 8 represents the effects of each parameter on the drain current. The most significant change in the simulation was clearly the channel length. Based on Equation (13), with the increasing channel length, the resistance in the channel was increased due to channel-length modulation.

Table 7 shows that the SNR responses for the highest current output for the FET were produced by Minitab-16. The results indicate the optimal levels of the control factors for both dependent parameters. The decision of the factor rank on the response table is also needed to determine which factor will be involved in the decision to predict the optimum condition value. These results are concurrent with the graphical form in Figure 8. From these graphs, the optimal condition value for increasing the current output can be easily obtained. This shows that the highest current output was produced when the channel length was 3 μm and the electrode size was 3 μm × 20 µm, with a thickness of 50 nm for the Ag electrode. These are the optimal conditions for the device.

An analysis of variance (ANOVA) helps to understand the relative magnitude of factor effects, which considers the optimisation objective. An ANOVA was performed to investigate the influence of the process parameters on the output drain current, I_d_. The grand total sum of squares can be decomposed into the sum of squares due to the mean and total sum of squares. The sum of squares is reported in Table 8, which measures the relative importance of each factor.
(17)$$\mathrm{Total}\;\mathrm{sum}\;\mathrm{of}\;\mathrm{squares}=\sum_{i=1}^{18}(n_{i}-m{)}^{2}$$
where *m* is the average of all the *n_i_* values from Equation (16). 

Table 8 shows the results of the ANOVA. The *p*-value for channel length (*L_C_*) was less than 0.05, which showed the most significant factor. However, the *p*-values of the other parameters were less significant compared to the channel length. These results show that even a slight change in the channel length will affect the drain current, I_d_. The percentage of contribution was observed to discover the factor significantly affecting the SNR of the output drain current, I_d_. Based on Table 8, channel length contributed the highest percentage (63.11%), followed by electrode length (17.87%), electrode width (9.93%), thickness of the electrode (0.82%) and type of electrode (2.67%), at a 95% confidence level. The percentage of error contribution was 5.6%, which means that all the factors significantly affected the SNR. According to the Taguchi method, to conclude whether all the important factors affected the response variable, the percentage of error contribution must be less than 50% [42]. 

Figure 9 shows the standardised residuals in normal probability plots in Taguchi analysis in response to the SNRs. The data plot helps to assess the quality of the model and its ability to accurately fit the data. Figure 9 shows that the data did not deviate too far from a straight line, indicating that the simulation design model is acceptable.

### 3.2. Confirmation Test

Using the ideal geometrical structures suggested by the research, two confirmation experiments were conducted. However, after completing the fabrication process, Au was selected as an electrode material rather than Ag because of its greater reliability and long-term stability. Despite having a better conductivity than Au, Ag has a lower melting point than Au, making it impractical for annealing at high temperatures. The first experiment compared the optimum conditions in the simulation and the experiment. Raman analysis was conducted to confirm the presence of the TiO_2_-rGO material on the device.

#### 3.2.1. Electrical Measurement

The current–voltage (I–V) measurements from the simulation and experimental measurements are shown in Figure 10 for the parameters under optimal conditions at a gate voltage (V_g_) of 3 V. The drain voltage (V_d_) was swept from 0 V to 0.5 V. Both graphs indicate a similar curve trend of n-type material behaviour due to a higher concentration of TiO_2_ in the nanocomposites. The curve difference between each graph was calculated at 28.7%. The are several factors that could be identified as the potential source for the 28.7% deviation. This could include the variations in the experimental conditions, some undefined conditions/parameters in the simulation model or the process of material synthesis.

#### 3.2.2. Raman Analysis

The presence of TiO_2_ and rGO in the thin film was investigated using a Raman spectrometer (DXR Raman microscope). The Raman spectra of the material are shown in Figure 11. The characteristics of the TiO_2_ spectrum peak for anatase phase showed 4 peaks at 138.69 cm^−1^, 393.24 cm^−1^, 518.59 cm^−1^ and 636.22 cm^−1^ for E_g_, B_1g_, A_1g_ and E_g_, respectively. The rGO spectrum showed 2 peaks at 1384.47 cm^−1^ due to the sp^3^ defects (D band) and a G band at 1617.81 cm^−1^, which can be assigned to the in-plane vibrations of sp^3^ carbon atoms [43]. The second order of the zone boundary phononsor 2D band, which relates to the stacking nature of graphene layers, was not observed. This was because the peak broadened for the TiO_2_-rGO nanocomposite as a consequence of multilayer rGO stacking by the decreasing the functional groups attached to the rGO [43]. 

TiO_2_ rutile has a higher sensitivity due to its smaller grain size and larger active surface area. However, the small bandgap in rutile makes it more conductive, making it unsuitable as a transducer material for biosensor application. rGO also has the properties of a large active surface area for functionalisation. Adding rGO to TiO_2_ anatase can reduce the bandgap and increase the active surface area, which increases the sensitivity of the biosensor [29].

## 4. Future Works and Challenges

In designing a high-performance FET biosensor, the device must be able to generate a high drain current, I_d_. However, some limitations must be considered. Although a smaller gap (*L_C_*) yields a higher conductivity, while also contributing to a lower detection limit, fabricating a smaller gap of less than 3 µm is a major challenge. We need to use a chrome mask to pattern a smaller channel gap of less than 20 µm and with higher photolithography resolution. In simulations, a thinner electrode of less than 50 nm can produce a higher I_d_ compared to a thicker electrode. However, the electrode of less than 50 nm-thick easily detaches during fabrication and measurement. The TiO_2_-rGO material cannot conduct a current in the dry state, and the sensitivity test using DI water and tap water only provided a general view of the detection limit for the device.

In the future, the surface of the FET transducer material will be functionalised with bioreceptors to improve the selectivity and sensitivity and enable the detection of specific biomolecules or analytes in complex biological samples.

## 5. Conclusions

The optimisation of the geometrical parameter for the TiO_2_-rGO biosensor was performed by applying the Taguchi method in the charge transport solver using the finite element method. In the application of the orthogonal array L_18_ (6^1^3^4^) mixed-level method to study the cause of factors on the variable behaviour, five factors with six and three levels were chosen: channel length (*L_C_*), electrode length (*L*), electrode width (*W*), thickness of electrode (*T*) and type of electrode (*C*). A series of simulations were carried out using the finite element method based on Taguchi’s experimental designs. According to the ANOVA results, the most significant factor was L_C_, which contributed the highest percentage, at 63.11%, with a *p*-value of less than 0.05. 

The optimal condition for the highest I_d_ was when *L_C_* was at 3 µm, with an electrode size of 3 µm × 20 µm, an electrode thickness of 50 nm and with silver (Ag) as the type of electrode. However, after completing the fabrication process, Au was selected as the electrode material rather than Ag because of its greater reliability and long-term stability. Despite having a better conductivity than Au, Ag has a lower melting point, making it impractical for annealing at high temperatures. The I–V measurement from both the simulation and experiment under optimal conditions showed a similar trend, and the curve difference between each graph was calculated at 28.7%. Raman analyses were conducted to validate the quality of TiO_2_-rGO.

As a result, the reliability of the Taguchi method as an effective simulation-based strategy in the design of numerical simulations to study the higher drain current was presented. This study can provide experts and researchers with the tools to obtain the ‘closest first assumption’, while at the same time avoiding the complexity of fabrication prior to analyte detection using the FET biosensor.

## Figures and Tables

**Figure 1 micromachines-14-01664-f001:**
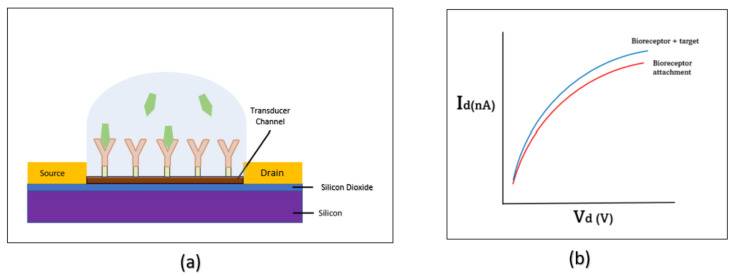
(**a**) Working principle of the FET biosensor. (**b**) I–V measurement of the FET biosensor.

**Figure 2 micromachines-14-01664-f002:**
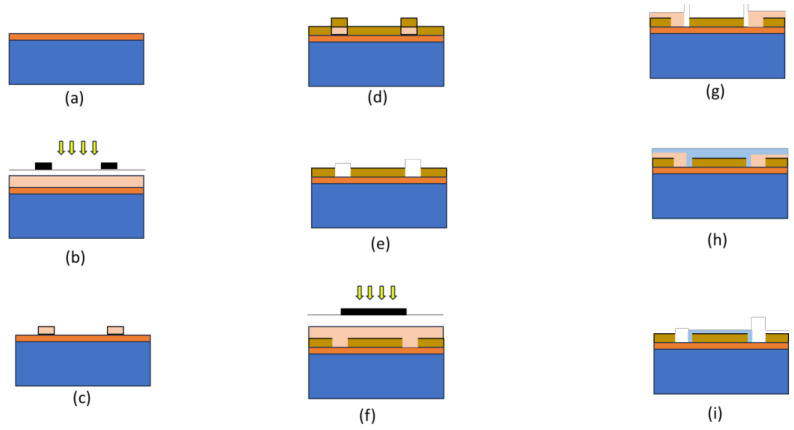
Illustration of the fabrication flow ((**a**–**i**) in sequence) of the electrode: (**a**) a cleaned p-type silicon wafer, (**b**) application of photoresist and UV exposure on the exposed surface for patterning, (**c**) development of photoresist, (**d**) deposition of Cr/Au using the electron beam, (**e**) removal of photoresist and lift-off, (**f**) second mask application for patterning, (**g**) development of photoresist, (**h**) deposition of TEOS and (**i**) removal of photoresist.

**Figure 3 micromachines-14-01664-f003:**
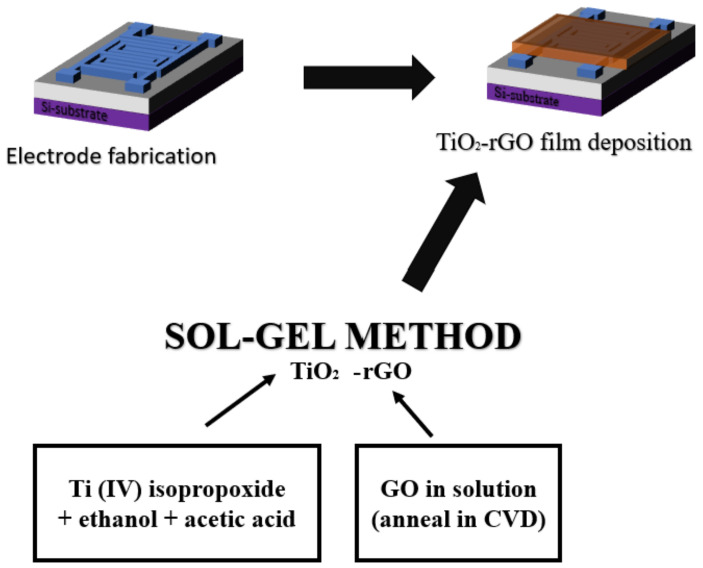
Illustration of the material synthesis and deposition process.

**Figure 4 micromachines-14-01664-f004:**
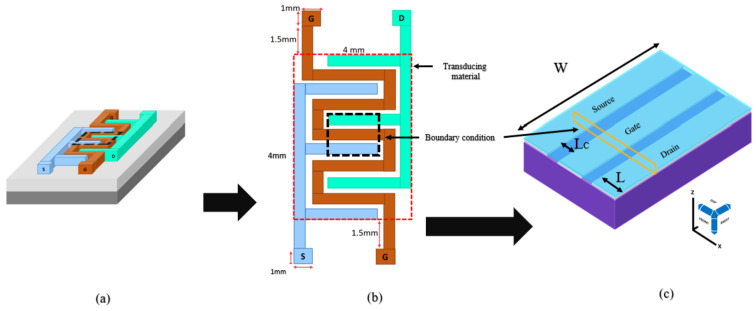
Pictorial representation of the simulated design: (**a**) schematic of the FET on a wafer with the source (S), gate (G) and drain (D), (**b**) schematic design from the top view and (**c**) design simulation in CTS for electrical characterisation with the electrode width (W) at the X, Y, Z view.

**Figure 5 micromachines-14-01664-f005:**
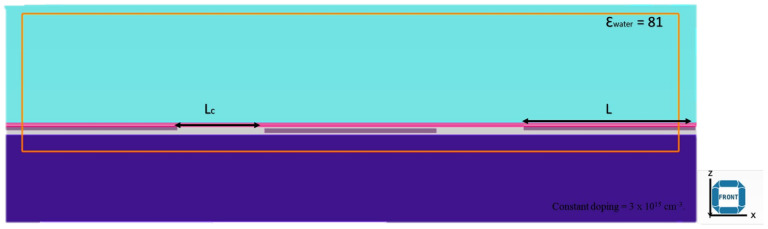
Schematic of the FET from the X, Z view for electrical characterisation, with channel length (L_C_) and electrode length (L).

**Figure 6 micromachines-14-01664-f006:**
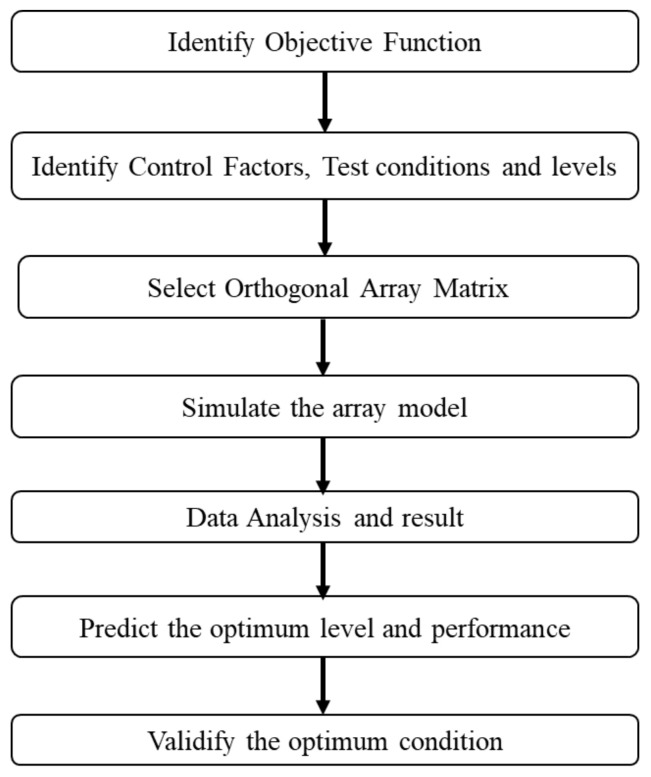
Flow chart of the Taguchi method.

**Figure 7 micromachines-14-01664-f007:**
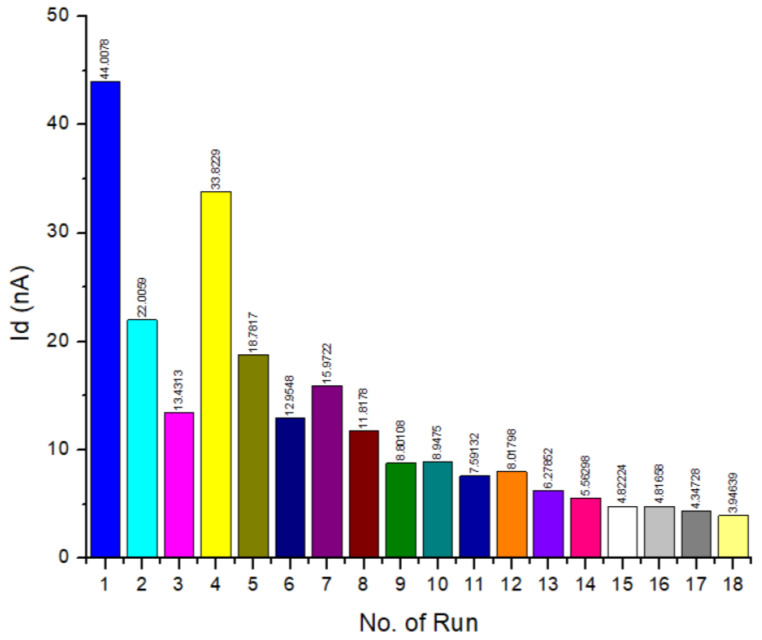
Drain current (I_d_) of all simulated designs.

**Figure 8 micromachines-14-01664-f008:**
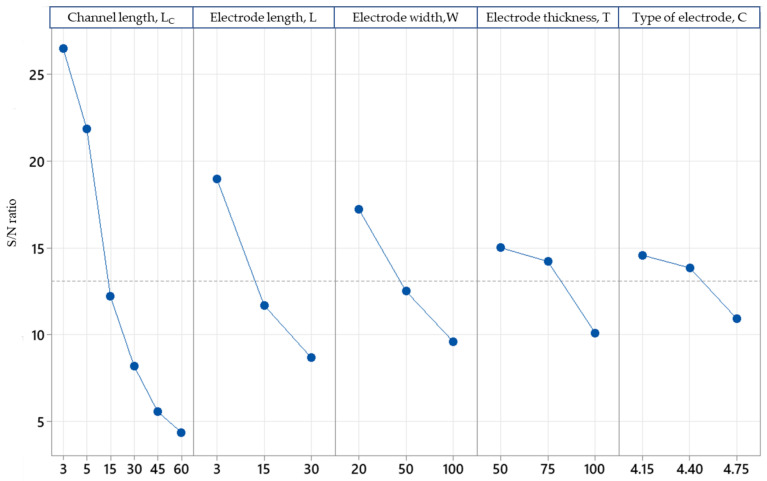
Plot of the main effects for the mean of the SNR of the drain current (I_d_) for optimisation of the TiO_2_-rGO FET.

**Figure 9 micromachines-14-01664-f009:**
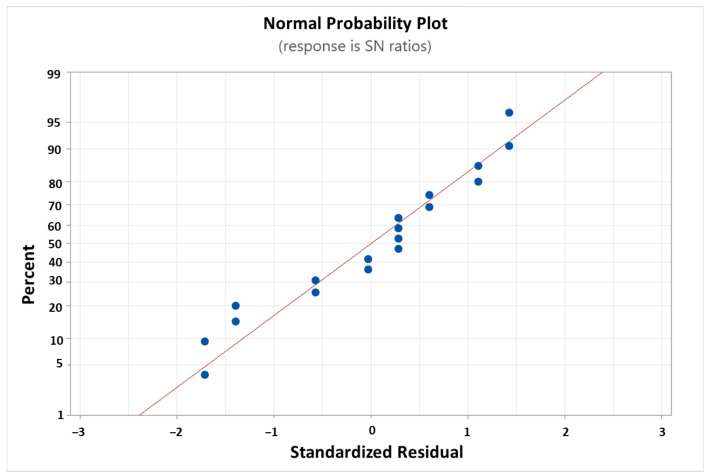
The standardised residuals in normal probability plots in response to SNRs.

**Figure 10 micromachines-14-01664-f010:**
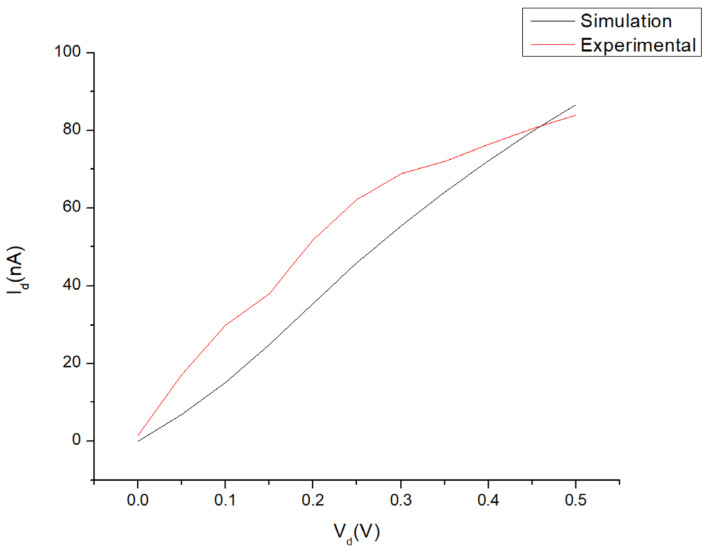
I–V graph of the FET device with TiO_2_-rGO deposition (labelled as experimental) compared to the simulation value (labelled as simulation).

**Figure 11 micromachines-14-01664-f011:**
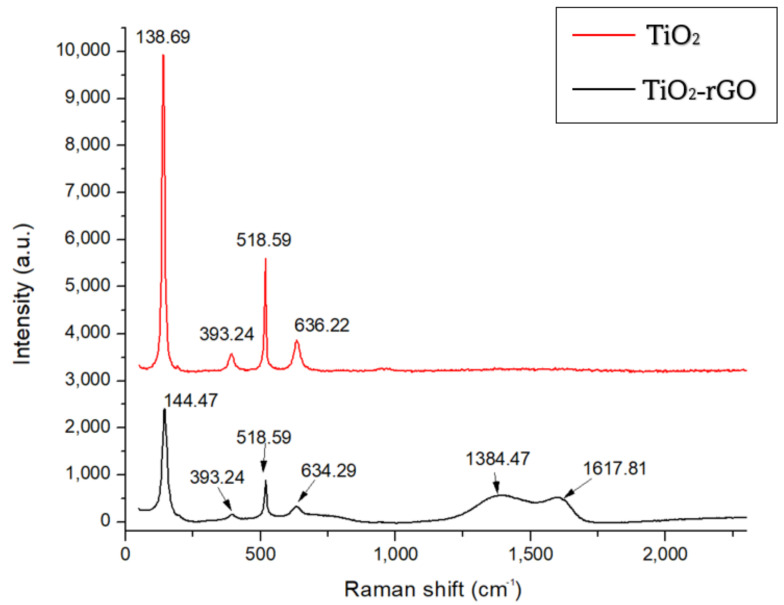
Raman spectra of TiO_2_ and the TiO_2_-rGO nanocomposite.

**Table 2 micromachines-14-01664-t002:** Important electrical parameters of the TiO_2_-rGO FET biosensor.

Parameters	Description	Nominal Values
$\mu_{n}$	Electron mobility	0.017 cm^2^ V^−1^s^−1^
$\mu_{p}$	Hole mobility	0.3 cm^2^ V^−1^s^−1^
$N_{s}$	Substrate concentration P doping	3 × 10^15^ cm^−3^
$\tau_{n}$	Electron SRH recombination lifetime	24 ns
$\tau_{p}$	Hole SRH recombination lifetime	24 ns
SRV	Surface recombination velocity	1 × 10^7^
$\varepsilon_{r}$	Relative permittivity for TiO_2_-rGO	3.5
$m_{n}^{*}$	Effective mass of electron for TiO_2_-rGO at 300 K	0.0949 × 1/me
$m_{h}^{*}$	Effective mass of hole for TiO_2_-rGO at 300 K	0.562 × 1/me
$E_{g}$	Bandgap	3.1 eV

**Table 3 micromachines-14-01664-t003:** Geometrical parameters of the FET.

Parameters	Description	Value Range
*L_C_*	Channel length	1–60 µm
*L*	Electrode length	3–30 µm
*W*	Electrode width	20–100 µm
*T*	Electrode thickness	50–100 nm
*C*	Type of electrode	Ag (4.15), Cu (4.4), Au (4.75)

**Table 4 micromachines-14-01664-t004:** Control factors and levels of the FET-based biosensor.

Factor	Level					
	1	2	3	4	5	6
*L_C_*, µm	3	5	15	30	45	60
*L*, µm	3	15	30			
*W*, µm	20	50	100			
*T*, nm	50	75	100			
*C*, *W_F_*	4.15	4.40	4.75			

**Table 5 micromachines-14-01664-t005:** Layout of L18 (6^1^3^4^).

RUN	Channel Length, *L_C_*	Electrode Length, *L*	Electrode Width, *W*	Electrode Thickness, *T*	Type of Electrode, *C*
1	3	3	20	50	4.15
2	3	15	50	75	4.4
3	3	30	100	100	4.75
4	5	3	20	75	4.4
5	5	15	50	100	4.75
6	5	30	100	50	4.15
7	15	3	50	50	4.75
8	15	15	100	75	4.15
9	15	30	20	100	4.4
10	30	3	100	100	4.4
11	30	15	20	50	4.75
12	30	30	50	75	4.15
13	45	3	50	100	4.15
14	45	15	100	50	4.4
15	45	30	20	75	4.75
16	60	3	100	75	4.75
17	60	15	20	100	4.15
18	60	30	50	50	4.4

**Table 6 micromachines-14-01664-t006:** L18 layout: drain current (I_d_) results of the FET-based biosensor.

RUN	Current I_d_ at 1 V, nA	RUN	Current I_d_ at 1 V, nA
1	44.0078	10	8.9475
2	22.0059	11	7.59132
3	13.4313	12	8.01798
4	33.8229	13	6.27852
5	18.7817	14	5.56298
6	12.9548	15	4.82224
7	15.9722	16	4.81658
8	11.8178	17	4.34728
9	8.80108	18	3.94639

**Table 7 micromachines-14-01664-t007:** Response table for the “larger-the-better” SNR of the FET biosensor.

Level	*L_C_*	*L*	*W*	*T*	*C*
1	27.43	22.69	21.06	20.6	20.56
2	26.1	19.84	20.39	20.71	20.36
3	21.47	17.9	18.98	19.11	19.51
4	18.24				
5	14.84				
6	12.78				
Delta	14.65	4.8	2.09	1.6	1.06
Rank	1	2	3	4	5

**Table 8 micromachines-14-01664-t008:** ANOVA for SNR for the FET biosensor.

Source	DF	SS	Adj. SS	Adj. MS	F-Value	*p*-Value	Contribution (%)
Channel length, µm	5	1241.39	1241.39	248.28	8.77	0.028	63.11
Electrode length, µm	2	337.22	337.22	168.61	5.95	0.063	17.87
Electrode width, µm	2	178.59	178.59	89.29	3.15	0.151	9.93
Thickness of electrode, nm	2	83.36	83.36	41.68	1.47	0.332	0.82
Type of electrode, W_F_	2	45.3	45.3	22.65	0.8	0.51	2.67
Residual error	4	113.26	113.26	28.31			5.6
Total	17	1999.12					100

## Data Availability

The data are available within the manuscript.

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
