# Peer review of "Geometrical Characterisation of TiO2-rGO Field-Effect Transistor as a Platform for Biosensing Applications"

_micromachines, 2023, doi:10.3390/mi14091664_

Round 1
Reviewer 1 Report
This manuscript presents a top-gate FET developed using the finite element method within a 3D simulation model. The FET incorporates a transducer material, a nanocomposite of titanium dioxide and reduced graphene oxide, which serves as a platform for biosensing applications. It was not surprising that the channel length plays the most significant role in Id among other factors. Please find the comments below.
1) The reviewer recommends clarifying the voltage applied for each terminal in Line 320. Is it the Idlin or Idsat?
2) The reviewer believes the equation in Line 200 should be for holes. Please correct the equation.
3) Please follow consistent formatting when writing the paragraphs. Please correct the format of the paragraph in Line 286 and subsequent ones.
Minor editing of English language required.
Author Response
Thank you for giving us the opportunity to submit revise draft of my manuscript Title “Geometrical characterisation of TiO2-rGO Field-effect Transistor as Platform for Biosensing Applications” to Micromachines. We appreciate the time and effort that you have dedicated for providing your valuable feedback on my manuscript. We are grateful for your insightful comments on my paper. We have been able to incorporate changes to reflect most of the suggestions provided by the reviewers. We have highlighted the changes within the manuscript.
Here is a point-by-point response to reviewer’s comments and concerns.
Please see the attachment

Reviewer 2 Report
This manuscript reports on the “Geometrical characterization of TiO2-rGO Field-effect Transistor as Platform for Biosensing Applications” The manuscript cannot be published in the present form due to the following issues:
1. The authors report a 28.7% difference between simulation and experimental curves under optimal conditions. It would be beneficial to delve deeper into the possible sources of this deviation and discuss the implications for the biosensor's real-world performance.
2. While the paper mentions Raman analyses to validate the quality of TiO2-rGO, the results and implications of this analysis could be elaborated upon. Discussing how the quality of the transducer material affects overall biosensor performance would enhance the paper's impact.
3. The paper could benefit from a more comprehensive discussion of the theoretical underpinnings of FET operation and its relevance to biosensing. This would aid readers less familiar with the subject matter to understand the research context.
4. Incorporating relevant graphs or figures depicting the simulation and experimental results, remarkably the comparison of curves under optimal conditions, would enhance the clarity and accessibility of the paper.
Author Response

(The authors gave the same response as above.)

Round 2
Reviewer 2 Report
As the comments have been responded to appropriately. I recommend the publication of the manuscript in its present form.